# The Impact of Mental Stress on Cardiovascular Health—Part II

**DOI:** 10.3390/jcm11154405

**Published:** 2022-07-28

**Authors:** Michael Y. Henein, Sergio Vancheri, Giovanni Longo, Federico Vancheri

**Affiliations:** 1Institute of Public Health and Clinical Medicine, Umea University, 90187 Umea, Sweden; michael.henein@umu.se; 2Brunel University, Middlesex, London UB8 3PH, UK; 3St. George’s University, London SW17 0RE, UK; 4Radiology Department, I.R.C.C.S. Policlinico San Matteo, 27100 Pavia, Italy; sergiovancheri@gmail.com; 5Cardiovascular and Interventional Department, S. Elia Hospital, 93100 Caltanissetta, Italy; giova.longo@gmail.com; 6Department of Internal Medicine, S. Elia Hospital, 93100 Caltanissetta, Italy

**Keywords:** mental stress, endothelial dysfunction, systemic inflammation, hypertension, arrhythmia, Takotsubo

## Abstract

Endothelial dysfunction is one of the earliest manifestations of atherosclerosis, contributing to its development and progression. Mental stress induces endothelial dysfunction through increased activity of the sympathetic nervous system, release of corticotropin-releasing hormone from the hypothalamus, inhibition of nitric oxide (NO) synthesis by cortisol, and increased levels of pro-inflammatory cytokines. Mental-stress-induced increased output of the sympathetic nervous system and concomitant withdrawal of the parasympathetic inflammatory reflex results in systemic inflammation and activation of a neural–hematopoietic–arterial axis. This includes the brainstem and subcortical regions network, bone marrow activation, release of leukocytes into the circulation and their migration to the arterial wall and atherosclerotic plaques. Low-grade, sterile inflammation is involved in all steps of atherogenesis, from coronary plaque formation to destabilisation and rupture. Increased sympathetic tone may cause arterial smooth-muscle-cell proliferation, resulting in vascular hypertrophy, thus contributing to the development of hypertension. Emotional events also cause instability of cardiac repolarisation due to brain lateralised imbalance of cardiac autonomic nervous stimulation, which may lead to asymmetric repolarisation and arrhythmia. Acute emotional stress can also provoke severe catecholamine release, leading to direct myocyte injury due to calcium overload, known as myocytolysis, coronary microvascular vasoconstriction, and an increase in left ventricular afterload. These changes can trigger a heart failure syndrome mimicking acute myocardial infarction, characterised by transient left ventricular dysfunction and apical ballooning, known as stress (Takotsubo) cardiomyopathy. Women are more prone than men to develop mental-stress-induced myocardial ischemia (MSIMI), probably reflecting gender differences in brain activation patterns during mental stress. Although guidelines on CV prevention recognise psychosocial factors as risk modifiers to improve risk prediction and decision making, the evidence that their assessment and treatment will prevent CAD needs further evaluation.

## 1. Introduction

Epidemiological studies suggest that acute and chronic mental stress, such as anger, fear, job strain, depression or exposure to natural disasters, may contribute to the pathogenesis and development of coronary artery disease (CAD) [1,2,3,4]. The first part of the review on the relationship between psychological stress and cardiovascular (CV) disease focused on the brain response to negative emotions and autonomic nervous system imbalance, leading to hemodynamic and coronary artery responses to mental stress [5]. The aim of Part II is to review the evidence for the clinical association between mental stress and systemic CV effects, including endothelial dysfunction, inflammation, hypertension, arrhythmia, and stress cardiomyopathy, based on most recent literature findings.

## 2. Recognised Underlying Mechanisms That Can Be Measured in Laboratory

***Effects of mental stress on vascular endothelium*****.** The functional integrity of the vascular endothelium has a critical role in transducing mental stress into coronary artery disease (CAD). The endothelium lines the vascular inner wall and regulates homeostatic functions such as vascular tone, permeability, proliferation of vascular smooth muscle cells, platelet aggregation, and balance between thrombogenesis and fibrinolysis [6,7,8]. It is the main factor regulating coronary blood flow. Since myocardial oxygen extraction is almost maximal at rest, increases in myocardial metabolic activity must be balanced by proportional increases in coronary blood flow. Flow resistance is inversely proportional to the fourth power of the radius; hence, the distribution of coronary blood flow is strictly dependent on the arterial diameter. Under resting conditions, the endothelium maintains high vascular tone [9]. Under physiological stimuli, such as exercise, the endothelial cells modulate an appropriate dilatation of coronary arteries by locally releasing nitric oxide (NO), thus allowing the increase in coronary blood flow [10]. NO also has anti-inflammatory properties which protect the integrity of the endothelium by inhibiting fibrosis, platelet aggregation, and apoptosis [11,12].

The endothelium releases endothelium-derived contracting factor, endothelin-1 (ET-1), which mediates the vasoconstriction response to noradrenaline, particularly in the microvascular bed, balancing vasodilation with vasoconstriction. Clinical studies have shown that public speaking and anger provocation are associated with increased endothelial-cell-derived microparticles, which derive from the membranes of apoptotic endothelial cells and serve as a marker of endothelial damage [13,14].

Endothelial dysfunction is characterised by reduced bioavailability of NO and increased endothelin, leading to an imbalance between vasodilatation and vasoconstriction. Endothelial dysfunction can be identified by measuring flow-mediated dilatation (FMD). During brachial FMD testing, endothelium-dependent vasodilatation is induced by reactive hyperaemia, due to an increase in endothelial shear stress [15]. The underlying mechanisms for mental-stress-induced endothelial dysfunction include (a) increased activity of the sympathetic nervous system, which induces endothelial dysfunction by direct stimulation of α-adrenergic receptors, through noradrenaline released in the arterial walls; (b) the release of corticotropin-releasing hormone from the hypothalamus, which occurs within seconds following exposure to stress, contributing to endothelial dysfunction by stimulating the release of the vasoconstrictor ET-1 from the endothelium; (c) release of cortisol, 20–40 min after the onset of stress, which impairs endothelial function by inhibiting NO synthesis and increasing ET-1 release; (d) increased levels of pro-inflammatory cytokines, such as interleukins and tumour necrosis factor alpha, that directly impair endothelial function [11,14,16,17,18]. Additionally, mental stress may act in combination with the conventional cardiovascular risk factors to accelerate the development of atherosclerosis [19]. This may explain the limited effect of negative emotions on atherosclerosis in young age.

The immediate response to acute mental stress depends on the activity of the sympathetic nervous system, while the delayed response is regulated by release of cortisol and ET-1 [16]. In healthy subjects without clinical evidence of CAD, a brief episode of mental stress can induce a rapid impairment of endothelium-dependent vasodilatation that may last for hours, even after resolution of the hemodynamic changes [16,20]. This suggests that transient, repeated stress, as encountered during daily life, may result in clinically relevant effects, such as atherosclerosis and MSIMI [21]. Moreover, individuals with clinical depression have impaired endothelial function, which may account for their increased risk of CAD [22].

Endothelial dysfunction is one of the earliest manifestations of atherosclerosis, often preceding angiographic evidence of atherosclerotic plaque or increased intima-media ratio on ultrasound examination, contributing to its development and progression [23,24]. In healthy endothelium, dilatation occurs in response to changes in localised blood flow, shear stress or increased resistance. Impairment of endothelial control of vascular tone in small resistance arteries, which are responsible for most of the total peripheral resistance, can greatly affect tissue perfusion and blood pressure, and can be critical for myocardial perfusion. In addition, endothelial dysfunction induces activation of macrophages, proliferation of vascular smooth muscle cells, and platelet aggregation and adhesion to the walls of blood vessels, leading to both chronic stable myocardial ischemia and acute ischemic events [10].

Impairment of endothelium-derived vasodilatation may simultaneously occur in coronary microcirculation, epicardial arteries, and peripheral arterial systems, thus representing a systemic arterial disorder [24,25]. However, a segmental pattern along the coronary arteries has also been demonstrated [26]. In these patients, coronary segments with abnormal endothelial function are associated with lipid-rich plaques and necrotic core areas, which contain sites of active inflammation, microcalcifications and expression of vulnerable plaque. Inflammation may be a link between endothelial dysfunction and plaque composition [27,28]. Serial intravascular ultrasound evaluations have found accelerated progression of atherosclerosis in coronary segments with endothelial dysfunction [29].

***Inflammatory response to mental stress.*** Mental-stress-induced sympathetic activation drives vascular inflammation and promotes progression of atherosclerosis, plaque destabilisation, and increased risk of developing heart failure [30,31,32,33,34]. Systemic inflammation is involved in all steps of atherogenesis, from plaque formation to rupture [35,36]. Chronic stress may contribute to atherosclerosis by inducing systemic, sterile, low-grade inflammation, different from the response to acute infections [37]. Some imaging studies have shown that the activity of the amygdala is increased under conditions of both acute or chronic stress [34,38,39]. The amygdala regulates the stress perception and emotional response of the central nervous system network associated with carotid and coronary arterial inflammation, and hence it is implicated in atherosclerosis progression and plaque destabilisation [40,41].

In normal conditions the vagal (parasympathetic) efferent innervations release acetylcholine both in the reticuloendothelial system and in the heart, blocking the release of inflammatory cytokines, including interleukins (IL-1, IL-2, IL-6) and tumour necrosis factor alpha (TNF-α), thus modulating the inflammatory response to acute inflammation. This cholinergic anti-inflammatory pathway is also termed “parasympathetic inflammatory reflex” [42,43,44,45]. Mental stress induces an imbalance in autonomic regulation, resulting in increased output of the sympathetic nervous system and concomitant parasympathetic (vagal) nervous system withdrawal. Thus, reduced parasympathetic activity with mental stress results in the release of inflammatory cytokines by immunocompetent cells expressing acetylcholine receptors, including myocardial vasculature macrophages [46,47,48]. In turn, TNF-α can provoke the release of the vasoconstrictor ET-1 from activated macrophages, thereby promoting vasoconstriction and endothelial dysfunction, leading to coronary plaque rupture with clinical consequences [49]. The inflammatory response is also characterised by increased levels of C-reactive protein (CRP), an acute phase reactant used as a sensitive marker of systemic inflammation, strongly associated with the risk of CAD, ischemic stroke, and vascular mortality [50,51,52,53].

Mental stress also promotes the activation of a neural–hematopoietic–arterial axis, which includes the brain stress network, bone marrow, and arterial inflammation (Figure 1) [38,54,55]. The mental-stress-induced release of noradrenaline promotes hematopoietic stem cell proliferation in the bone marrow, leading to an increased release of leukocytes, in particular neutrophils and monocytes, into the circulation [56]. In conditions of chronic inflammation, high catecholamine levels induce pro-inflammatory changes in monocytes [57]. These inflammatory cells migrate to the arterial wall and infiltrate the atherosclerotic plaques, where they induce inflammation [58]. Proteases released from inflammatory leukocytes weaken the plaque fibrous cap, which may lead to plaque rupture and acute coronary events [59,60,61]. The importance of bone marrow activation and arterial inflammation in the development of atherosclerosis is confirmed by the observation that higher blood levels of monocytes and neutrophils correlate with increased CAD incidence and mortality [62].

Inflammatory mediators induced by chronic stress, in particular IL-6, activate the HPA axis, which is involved in the containment of inflammatory reactions, resulting in cortisol secretion by the adrenal gland [63]. However, the mechanisms by which cortisol influences atherosclerosis are not well defined. Although cortisol has anti-inflammatory actions, chronic exposure to elevated levels may induce a state of resistance, reducing the anti-inflammatory actions while increasing the concentration of inflammatory cytokines [64]. This effect may account for the association between increased circulating cortisol levels and increased perivascular inflammation, endothelial dysfunction, and progression of coronary artery calcification [18,65,66]. However, prolonged activation of the HPA axis due to chronic stress or repeated episodes of acute stress may lead to its dysfunction, resulting in reduced anti-inflammatory activity of cortisol and progression of the inflammatory response in the arterial wall [67,68].

Mental stress activation of the sympathetic nervous system also induces platelet activation. Activated platelets release inflammatory cytokines and adhesive molecules, which induce leukocyte recruitment on the endothelial surface, resulting in increased coagulability and a prothrombotic state. In turn, these conditions enhance the recruitment of leukocytes and platelets on the vessel wall, promoting the development of arterial thrombosis [31,69].

The central role of inflammation in the pathogenesis and clinical manifestation of atherosclerosis has also been confirmed by the reduction in coronary events by colchicine, an anti-inflammatory drug that reduces the inflammatory activity of leukocytes [70,71]. Moreover, in stable patients with previous myocardial infarction and persistent systemic inflammatory response, blocking the interleukin-6 activity directly using the monoclonal antibody canakinumab resulted in significant reduction in high-sensitivity CRP levels and incidence of subsequent CV events, without changes in low-density lipoprotein cholesterol [72].

## 3. Clinical Conditions Associated with Mental Stress

***Hypertension*****.** Epidemiological studies provide support for the importance of psychological stress in the development of essential hypertension, although this relationship is still debatable [73,74,75,76,77,78,79]. Mental-stress-induced autonomic imbalance, with increased sympathetic tone accompanied by reduced parasympathetic tone, may cause smooth-muscle-cell proliferation and vascular remodelling, resulting in vasoconstriction [80]. Cortisol stress reactivity, an index of HPA axis function, is another possible mechanism through which mental stress may influence the risk of hypertension [81]. Longitudinal studies have shown that greater cardiovascular (CV) responses to stressful tasks are associated with long-term development of hypertension and with increased risk of cardiovascular mortality, independently of conventional CV risk factors [82,83,84,85,86]. The repeated episodes of CV activation result in vascular hypertrophy, leading to a progressive increase in peripheral resistance and thus contributing to the development of established hypertension. Moreover, increased systolic blood pressure reactivity to mental stress is positively associated with progression of atherosclerosis, as expressed by carotid intima-media thickness (IMT), an early marker of generalised atherosclerosis, including coronary arteries, strongly associated with CAD [87,88].

***Arrhythmia*****.** Both acute and chronic stressful events increase the risk of arrhythmia and sudden cardiac death (SCD) [89,90,91]. Ventricular arrhythmias in patients fitted with an implantable cardioverter-defibrillator (ICD) are 5 times more frequent after heightened anger than in control periods [92]. Anger and negative emotions are also associated with the initiation and progression of atrial fibrillation after adjusting for conventional CV risk factors, previous CAD, and valvar heart disease [93,94,95]. In individuals with inherited arrhythmogenic disorders characterised by congenital myocardial ion channelopathies, such as long QT syndrome (LQTS) and catecholaminergic polymorphic ventricular tachycardia (CPVT), emotional stress, as well as physical exercise, can induce potentially lethal ventricular tachyarrhythmias, in the absence of structural heart disease [96,97,98,99,100,101]. Cardiac autonomic dysfunction, characterised by enhanced sympathetic activity and parasympathetic withdrawal, is the link between mental stress and arrhythmia. Emotional factors may influence the risk of arrhythmia by inducing electric heterogeneities of ventricular depolarisation or repolarisation [102,103]. The myocardial ion channels are responsible for the transmembrane flow of sodium, potassium and calcium ions that determine the cardiac action potential and intracellular calcium homeostasis. In normal hearts, sympathetic activation shortens the action potential as the heart rate increases. In LQTS, a defective cardiac potassium channel results in a paradoxical increase in the QT interval due to pathological prolongation of the plateau phase of the action potential, leading to abnormal depolarisations during the repolarisation phase of the action potential, defined as early afterdepolarisations (EADs), which can cause lethal arrhythmias [104,105].

The genetic mutation in CPVT alters intracellular calcium handling. In addition to normal calcium release during systole, which underlies cardiac contraction, there is a spontaneous calcium release during diastole. As a consequence, sympathetic activation results in intracellular calcium overload, inducing oscillations of membrane potentials occurring after full repolarisation, known as delayed afterdepolarisations (DADs), which can trigger ventricular arrhythmias [106,107].

Beta-blockers (BB), especially non-selective nadolol and propranolol, are a first-line therapy for patients with a clinical diagnosis of LQTS or CPVT, with proven effects on arrhythmia and mortality reduction [108,109]. In patients with LQTS, in addition to BB, an implantable cardioverter-defibrillator (ICD) should be considered for those who experienced syncope or cardiac arrest. When BB and/or ICD are not effective or contraindicated, surgical left cardiac sympathetic denervation that reduces the catecholaminergic effect on left ventricular myocardium may be considered [96]. In patients with CPVT, in addition to BB and ICD, flecainide, alone or with BB, prevents ventricular arrhythmias. Patients experiencing episodes of multiple ventricular tachycardia/fibrillation over a short period of time (arrhythmic storm) may be treated with stellate ganglion blockade, using percutaneously infiltrating local anaesthetic agents [110].

Heterogeneity of repolarisation is also important in patients with heterogeneous cardiac denervation due to myocardial infarction or cardiomyopathies, inducing repolarisation instability, reflected in ECG T wave alternans, thereby increasing the risk of ventricular arrhythmia [91,111]. One suggested mechanism by which mental stress triggers arrhythmia is the “brain-heart laterality”, in which lateralised imbalance of autonomic neural stimulation of the heart may induce asymmetric repolarisation [112,113]. While the left sided sympathetic nerves are mainly distributed over the postero-inferior wall of the ventricles, the right sided are over the anterior wall, although with large interconnections. There is also anatomical evidence to support subcortical lateralisation of efferent sympathetic pathways, maintained from brainstem to peripheral nerves. Neuroimaging studies suggest that cortical regions in the right hemisphere, in particular right and ventral posterior insula subregions, exert sympathetic control [114,115]. Lateralisation of the processing of emotion in the brain and of autonomic inputs to the surface of the heart contribute to the repolarisation instability, which may lead to arrhythmia [116,117].

***Takotsubo cardiomyopathy*****.** Acute emotional stress may trigger a heart failure syndrome clinically presenting as acute ST elevation myocardial infarction, characterised by reversible left ventricular balloon-like wall motion abnormalities, usually involving the LV apex and rarely the mid ventricle, without significant coronary artery obstruction, termed Takotsubo syndrome (TTS) or “*broken heart syndrome*” (Figure 2 and Figure 3) [118,119,120]. The underlying cause is assumed to be an overstimulation of the sympathetic nervous system, triggered by emotional and physical stress, although, in a small proportion of patients, the syndrome is not directly related to any stressful event [121,122,123]. TTS shares common clinical presentation and pathophysiological mechanisms with “neurogenic stress cardiomyopathy”, which includes subarachnoid haemorrhage, stroke, either haemorrhagic or ischemic, traumatic brain injury, epilepsy, and central nervous system infections [124]. Depression and chronic anxiety disorders are frequent in TTS patients [125]. In addition to negative emotions, TTS can be rarely triggered by positive life events, including important personal achievements, marriage, and holidays. This condition has been described as “*happy heart syndrome*”. It is associated with clinical and laboratory findings similar to the “*broken heart syndrome*” [126,127]. The possible mechanism involved is central and subcortical brain areas, and the sympathetic activation induced by positive emotional excitement [128]. Despite being previously believed to be a transient benign disease, the syndrome carries high rates of complications, similar to negative outcomes of acute coronary events [122]. The higher prevalence of TTS in postmenopausal women seems to indicate that oestrogen deficiency may have a pathogenic role. Under physiological conditions, oestrogen has beneficial effects on coronary microcirculation tone and endothelial function, reducing the sympathetic response to mental stress and improving coronary blood flow [129,130,131]. During menopause, both increased sympathetic vasoconstriction and endothelial dysfunction are a consequence of reduced oestrogen levels [132].

Stress-induced cardiac sympathetic hyperactivity leads to an acute increase in myocardial and circulating catecholamine levels that induce direct myocyte injury likely due to calcium overload, coronary microvascular vasoconstriction, increased ventricular afterload resulting in oxygen supply–demand mismatch, endothelial dysfunction, and myocardial inflammation [121,133,134,135,136]. Histological alterations are typical of catecholamine toxicity, which consists of focal mononuclear inflammatory cells infiltration, a considerable increase in extracellular matrix protein levels, and contraction band necrosis (also known as myocytolysis). Neuroimaging studies have shown that patients with TTS have structural alterations in the limbic system and reduced functional connectivity among brain regions of the cortico-limbic network, comprising the prefrontal cortex, right insula, amygdala, hypothalamus, cingulate cortex, and thalamus, all of which regulate the autonomic nervous system and the CV response to mental stress [137,138,139,140,141]. The altered functional connectivity pattern may persist for weeks following the acute cardiac event, well after the recovery of left ventricular dysfunction [142]. Reduced functional brain connectivity may precede the development of TTS by years. A retrospective study of the alterations in neural connectivity in patients who subsequently developed TTS showed increased amygdalar activity years before developing TTS [143]. Such heightened amygdalar activity may predispose someone to TTS by potentiating the autonomic and neurohormonal response to future stressors.

## 4. Gender Differences in Response to Mental Stress

Women, especially young women, develop MSIMI more often than men [144,145,146]. There are gender differences in the brain activation pattern during mental stress [147,148]. Women with CAD have greater activation of cortico-limbic areas that regulate emotions and sympathetic response, including the amygdala, hippocampus, and prefrontal cortex, compared to men. This pattern of brain activation in women could translate into differences in neurohormonal and CV response to stress. Although men and women with CAD develop similar rate-pressure product response to mental stress, women have more marked endothelial dysfunction and intense vasoconstrictive response, resulting in microvascular dysfunction [144,149,150]. Microvascular dysfunction is common in women with chest pain, even in the absence of significant epicardial coronary obstruction [151]. This may be the substrate for MSIMI being more likely in women than in men.

## 5. Prevention and Treatment

Prevention strategies may be population-based (universal), involving the general population, or targeted to those at the highest risk of adverse events. Intervention studies of modifying stressors in healthy individuals are difficult to implement because they require changes in life habits, work organisation, and social support. In addition, stressors may be unpredictable and trigger CV events in very different conditions, such as family life, sport events and natural disasters. There are only few small general population randomised studies addressing this in the literature [152].

The rationale of focusing on high-CV-risk individuals is that given the low overall incidence rates of CV events in the healthy population, the reduction in absolute risk of CAD due to a reduction in stress-induced disease would be small. In contrast, in high-risk individuals, with high overall rates of CV events, the reduction in events due to stress would have larger effects [19]. In secondary prevention, cognitive behavioural therapy decreased the risk of recurrent CV events in patients with previous CAD and prolonged survival in women after acute myocardial infarction [153,154]. Treatment of depression in patients with recent myocardial infarction resulted in lower major coronary events compared to the placebo [155]. Treatment of major depression even in individuals without baseline CAD substantially reduced the development of coronary events [156]. In a small-size study of patients with stable CAD and MSIMI (see Part I [5]), treatment with escitalopram reduced the rate of MSIMI demonstrated on echocardiography and/or ECG, probably due to a beneficial effect on platelets aggregation [157]. However, other studies showed little or no evidence that treating depression in such patients improves CV outcomes [158,159]. A comprehensive systematic review of several studies of psychological treatments in patients with CAD, including those with and without a psychopathology at baseline, showed no effect on total mortality, myocardial revascularisation procedures and non-fatal myocardial infarction. However, there was low-quality evidence of a small reduction in cardiac mortality and improvement of psychological symptoms, such as depression, anxiety or stress [160]. Treatments have been more effective in patients with established psychological conditions at baseline. However, there was no impact on total mortality, risk of revascularisation procedures, or the rate of non-fatal myocardial infarction. In addition, most studies were of low quality, suggesting uncertainty about the results. Since exercise training is a crucial component of cardiac rehabilitation, and exercise has been shown to reduce depression symptoms in chronically ill patients, comprehensive cardiac rehabilitation including the psychological components should be recommended in patients with previous CAD [161,162]. Guidelines on CAD prevention recognise the relevance of psychosocial factors in the development of significant CAD. However, the evidence that their assessment and treatment may prevent CAD is still inconclusive [163,164].

## 6. Limitations

There are several limitations in the interpretation of the relationship between mental stress and CV disease. Firstly, the pathophysiological mechanisms linking mental stress to increased risk of CAD are complex and multifactorial, involving several variables, such as activation of brain circuits, the autonomic nervous system, the neuroendocrine and immune systems, conventional CV risk factors, and individual’s past experience. In turn, the resulting hemodynamic and inflammatory changes in their combined effects modulate the brain response to environmental stimuli [19,165]. These interactions may account for the large individual variability in the response to mental stress. This makes excluding confounding factors and identifying the role of mental stress in the development of CV disease difficult. Secondly, most available studies are observational and based on patient’s recalling of negative emotions preceding CV events. This method cannot determine whether the association between stress and CV event is causal [3]. Thirdly, there are large methodological and patient characteristic differences between studies [90]. Finally, some hemodynamic and vascular effects of mental stress may occur without symptoms or may develop long after the stressful event. Hence, the causal relationship cannot be precisely ascertained.

## 7. Conclusions

Over the past four decades, there has been an increasing awareness that mental stress is an important and potentially modifiable risk factor for acute and chronic CV events. Emerging epidemiological and experimental evidence suggest that stress-induced hemodynamic, vascular and inflammatory alterations may interact together and exert an important role in atherosclerosis progression, as well as in acute CV disease triggering, especially in high-risk individuals. However, in clinical practice, the assessment of the impact of mental stress on CAD disease is often missing. Despite current guidelines recognising the importance of mental stress as a CV risk factor, further research with large longitudinal studies is needed to define the prevention and treatment strategies required to reduce the CV effects of mental stress.

## Figures and Tables

**Figure 1 jcm-11-04405-f001:**
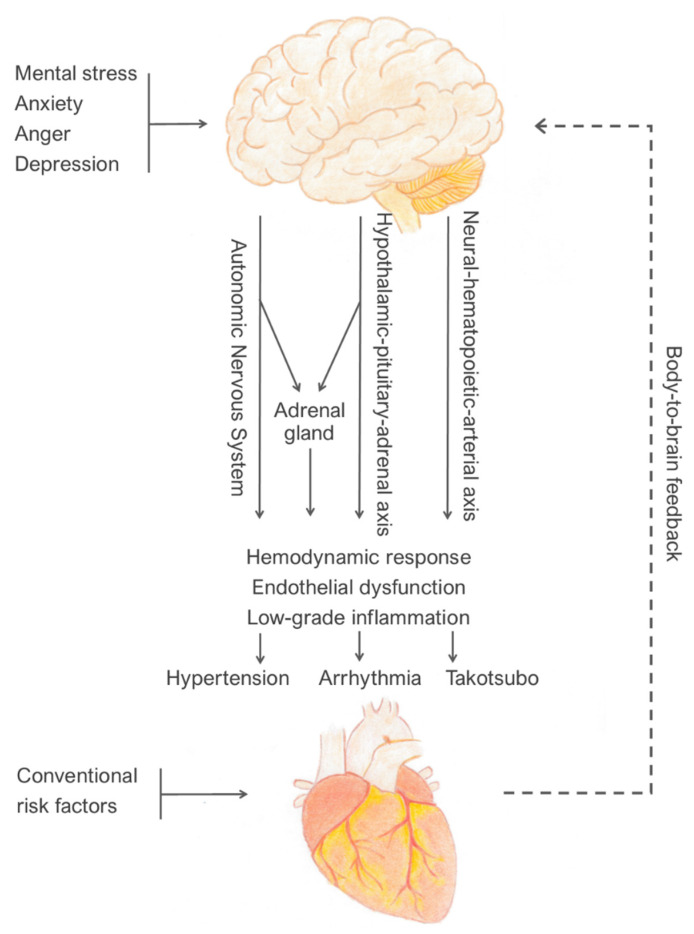
Schematic model of the mechanisms that translate negative emotion into CV disease. The brain response to mental stress involves activation of autonomic nervous system, hypothalamic–pituitary–adrenal axis, and neural–hematopoietic–arterial axis, resulting in pathophysiological effects that trigger CV events (brain-to-body). In turn, these effects modulate the brain response to stress (body-to-brain).

**Figure 2 jcm-11-04405-f002:**
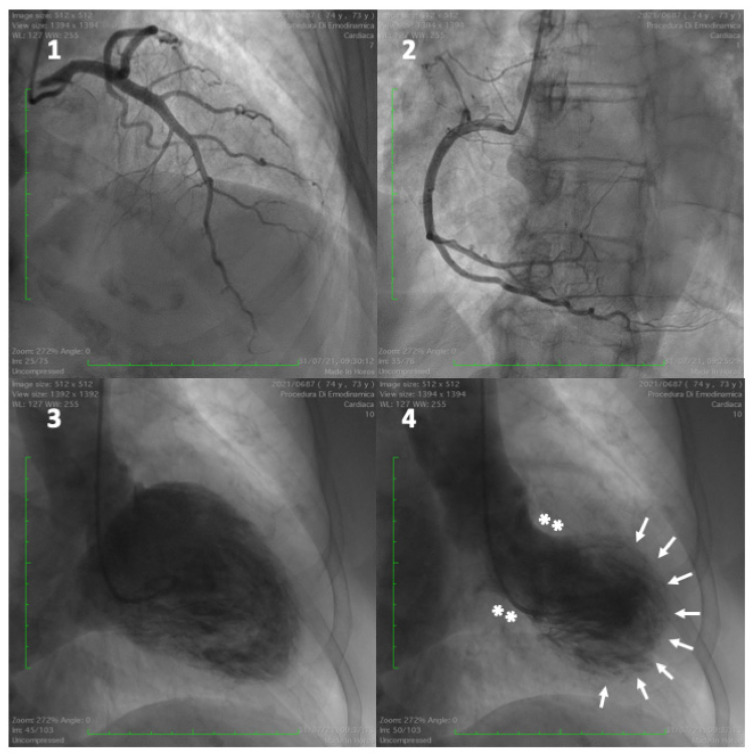
A 73-year-old woman admitted for anterior ST-elevation myocardial infarction. Coronary angiography showed no coronary lesions on left and right coronary arteries (frames (**1**) and (**2**)). Left ventriculogram (frames (**3**) and (**4**)) showed typical ballooning of the left ventricle apex (frame (**4**)), characterised by hypercontraction of the basal segments (with asterisks) and akinesia in the mid and apical segments (with arrows), typical for Takotsubo cardiomyopathy.

**Figure 3 jcm-11-04405-f003:**
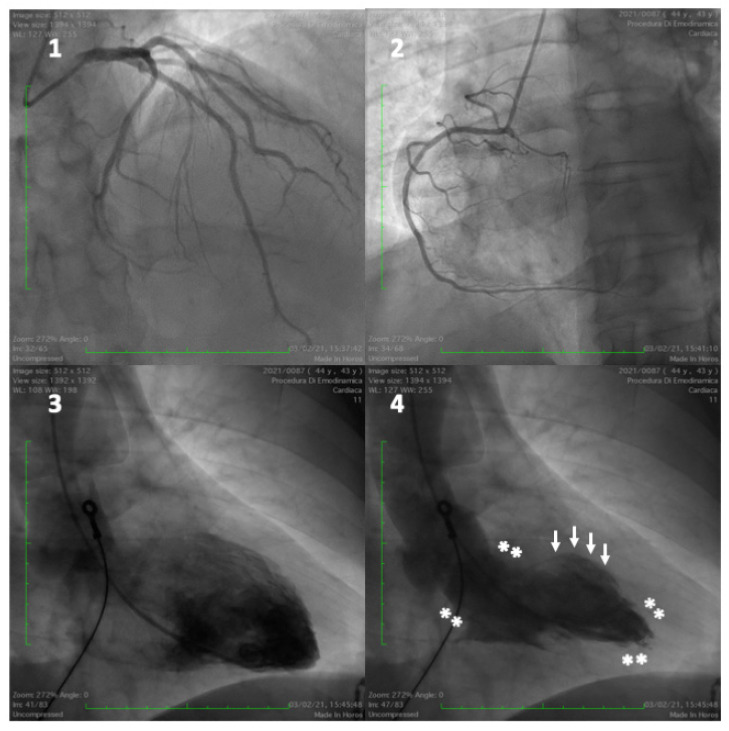
A 43-year-old man admitted for anterior ST-elevation myocardial infarction. Coronary angiography showed no coronary lesions on left and right coronary arteries (frames (**1**) and (**2**)). Left ventriculogram (frames (**3**) and (**4**)) showed hypercontraction of the basal segments and apex (frame (**4**), with asterisks) and akinesia in the mid segments (with arrows), typical for mid-ventricular Takotsubo cardiomyopathy.

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
