# Peer review of "The Impact of Mental Stress on Cardiovascular Health—Part II"

_jcm, 2022, doi:10.3390/jcm11154405_

Round 1

Reviewer 1 Report

Review articles: JCM 2022

This article is the second part of “The Impact of Mental Stress and Cardiovascular Health”. The part I of “The Impact of Mental Stress and Cardiovascular Health” has been published in June 10, 2022 in JCM, primarily focusing on mental stress induced myocardial ischemia along with other systematic responses to mental stress. The current article, the part II, aims at summarizing mental stress effects on endothelial and inflammatory functions, and three clinical entities, i.e. hypertension, arrhythmia, and Takotsubo cardiomyopathy. 

I sincerely applause the dedicated commitment and efforts of the authors in providing such thorough reviews upon literature primarily appeared in the new millennium. These rich data validates years of years speculations of clinicians, even folklore’s belief. Such reviews are welcome by medical professions and countless non-medical individuals.

The comment I would like to share is more of a writing style when presenting a review in such broad territories. As a Clinician-Scientist who tries to bridge the research to translational and more in basic field, I find that a systematic presentation with divisions into clinical presentations and underlying mechanisms are easier to capture and digest. A metaphor I like to use is “pilling the onion” to refer to this kind of presentation. For this particular article, given the part I has been published, the authors may organize it into something like; 1. Other clinical presentations or conditions associated with mental stress (HTN, arrhythmia, and Takotsubo cardiomyopathy), 2. Highly recognized underlying mechanisms that can be measured in laboratory, 3……

Author Response

Authors response. The writing style of the paragraphs has been modified according to the Reviewer suggestion:

  1. Recognized underlying mechanisms that can be measured in laboratory

Effects of mental stress on vascular endothelium.

Inflammatory response to mental stress.

  1. Clinical conditions associated with mental stress

Hypertension

Arrhythmia

Takotsubo cardiomyopathy

  1. Gender differences in response to mental stress
  2. Prevention and treatment

Limitations

Conclusions 

Reviewer 2 Report

Extremely interesting paper.

I only suggest to extend the paragraph about Takotsubo cardiomyopathy adding the recently recognized "happy heart" form.

Minor comments:

- MSIMI should be defined also in the text at first appearance

Author Response

Reviewer comment.

Extremely interesting paper.

I only suggest to extend the paragraph about Takotsubo cardiomyopathy adding the recently recognized "happy heart" form.

Authors response. Happy heart form of Takotsubo has been added to page 7: In addition to negative emotions, TTS can be rarely triggered by positive life events including important personal achievements, marriage, holidays. This condition has been described as “happy heart syndrome”. It is associated with clinical and laboratory findings similar to the “broken heart syndrome”. The possible mechanism involved is central and subcortical brain areas, and the sympathetic activation induced by positive emotional excitement.”

Minor comments:

- MSIMI should be defined also in the text at first appearance.

Authors response. First appearance of MSIMI definition is now in the Abstract

Reviewer 3 Report

In the study by Michael Y Henein, et al., they reviewed the association between mental health/mental stress and cardiovascular disease.

I feel this review is very well thought out and very cohesive.

However, there are some issues should be addressed appropriately.

I believe that arguing your insights about these issues and making a new version can add much value to your manuscript.

Major concern #1

You have shown a relation between mental or psychological stress and arteriosclerosis, but your interest is only in promoting factors, with no mention of suppressors.

 In general, marked arteriosclerosis does not occur in young generation, even if severe mental or psychological stress causes vascular endothelial damage.

Your review does not discuss this issue.

Major concern #2

I feel that the paragraph of arrhythmia is insufficient.

This is because mental stress is affected by autonomic regulation, and autonomic disorder can be closely related to the development of arrhythmias. Thus, clinically, it is more involved than other cardiovascular events.

Sympathetic nerve hyperactivity is involved in shortening and prolonging action potentials, enhancing abnormal trigger ability, and developing delayed after depolarization, and parasympathetic nerve hyperactivity affects cell membrane hyperpolarization and shortening of action potentials.

Whether an antiarrhythmic drug contains αβ receptor blockade and muscarinic receptor blockade influences the treatment choice.

Much deeper discussion sould be required about this issue; emotional and mental stress induce autonomic activity and arrhythmogenicity.

In addition, if we focus on emotion and sympathetic activity, we need to touch on the relationship with CPVT and LQT type 2.

Also, please comment on arrhythmia storm (sedation and sympathetic ganglion block are effective options).

Author Response

Reviewer comment.

In the study by Michael Y Henein, et al., they reviewed the association between mental health/mental stress and cardiovascular disease.

I feel this review is very well thought out and very cohesive.

However, there are some issues should be addressed appropriately.

I believe that arguing your insights about these issues and making a new version can add much value to your manuscript.

Major concern #1

You have shown a relation between mental or psychological stress and arteriosclerosis, but your interest is only in promoting factors, with no mention of suppressors.

Authors response. The mechanisms that regulate the central nervous system and autonomic response to psychological stress have been discussed in Part I, published in JCM few weeks ago, doi: 10.90/JCM11123353

 In general, marked arteriosclerosis does not occur in young generation, even if severe mental or psychological stress causes vascular endothelial damage.

Your review does not discuss this issue.

Authors response. This interesting point has been added to Page 3. “Also, mental stress may act in combination with the conventional cardiovascular risk factors to accelerate the development of atherosclerosis. This may explain the limited effect of negative emotions on atherosclerosis in young age.”

Major concern #2

I feel that the paragraph of arrhythmia is insufficient.

This is because mental stress is affected by autonomic regulation, and autonomic disorder can be closely related to the development of arrhythmias. Thus, clinically, it is more involved than other cardiovascular events.

Sympathetic nerve hyperactivity is involved in shortening and prolonging action potentials, enhancing abnormal trigger ability, and developing delayed after depolarization, and parasympathetic nerve hyperactivity affects cell membrane hyperpolarization and shortening of action potentials.

Whether an antiarrhythmic drug contains αβ receptor blockade and muscarinic receptor blockade influences the treatment choice.

Much deeper discussion should be required about this issue; emotional and mental stress induce autonomic activity and arrhythmogenicity.

In addition, if we focus on emotion and sympathetic activity, we need to touch on the relationship with CPVT and LQT type 2.

Authors response. Thank you for your suggestion. A paragraph has been added to page 6:

“In individuals with inherited arrhythmogenic disorders characterized by congenital myocardial ion channelopathies, such as long QT syndrome (LQTS) and catecholaminergic polymorphic ventricular tachycardia (CPVT), emotional stress, as well as physical exercise, can induce potentially lethal ventricular tachyarrhythmias, in the absence of structural heart disease.95-100 Cardiac autonomic dysfunction, characterized by enhanced sympathetic activity and parasympathetic withdrawal, is the link between mental stress and arrhythmia. Emotional factors may influence the risk of arrhythmia by inducing electric heterogeneities of ventricular depolarisation or repolarisation.101,102 The myocardial ion channels are responsible for the transmembrane flow of sodium, potassium and calcium ions that determine the cardiac action potential and intracellular calcium homeostasis. In normal hearts, sympathetic activation shortens the action potential as the heart rate increases.  In LQTS, a defective cardiac potassium channel results in a paradoxical increase in the QT interval due to pathological prolongation of the plateau phase of the action potential, leading to abnormal depolarisations during the repolarisation phase of the action potential, defined as early afterdepolarisations (EADs), which can cause lethal arrhythmias.103,104

The genetic mutation in CPVT alters the intracellular calcium handling. In addition to normal calcium release during systole, which underlies cardiac contraction, there is a spontaneous calcium release during diastole. As a consequence, sympathetic activation results in intracellular calcium overload, inducing oscillations of membrane potentials occurring after full repolarisation, known as delayed afterdepolarisations (DADs), which can trigger ventricular arrhythmias.105,106

Also, please comment on arrhythmia storm (sedation and sympathetic ganglion block are effective options).

Authors response. A paragraph about the antiarrhythmic treatment of channelopathies and arrhythmic storm has been added to page 6:

Beta-blockers (BB), especially non-selective nadolol and propranolol, are first-line therapy in patients with clinical diagnosis of LQTS or CPVT, with proven effects on arrhythmia and mortality reduction.108,109 In patients with LQTS, in addition to BB, implantable cardioverter-defibrillator (ICD) should be considered in those who experienced syncope or cardiac arrest. When BB and/or ICD are not effective or contraindicated, surgical left cardiac sympathetic denervation that reduces the catecholaminergic effect on left ventricular myocardium, may be considered.96 In patients with CPVT, in addition to BB and ICD, flecainide, alone or with BB, prevents ventricular arrhythmias. Patients experiencing episodes of multiple ventricular tachycardia/fibrillation over a short period of time (arrhythmic storm), may be treated with stellate ganglion blockade, using percutaneously infiltrating local anaesthetic agents.110

This manuscript is a resubmission of an earlier submission. The following is a list of the peer review reports and author responses from that submission.

Round 1

Reviewer 1 Report

This is a very informative and condenced description of the physiological mechanisms connecting mental stress with coronary heart disease and one particular cardiomyopathy (Takotsubo).

My major concern is that I do not understand the context or rational. The title suggests "Part II". Where is then part I? It is some kind of ambitious but super-condenced litterature review but the title is not informative and it doesnt follow the normal structure for a review. Nor is there any aim or any guiding text for the reader to help him/her understand the context or the rational behind this article.

To my understanding the content in this ms is overall correct. However, it is not obvious what the principles for selection of the content are. Some headings are misleading. E.g. the Introduction is not an introduction (an introduction would have been appreciated). Mental Stress-Induced Cardiomyopathy is not the same thing as Takotsubo which is suggested by one heading. Putting MSIMI together with prevention and treatment is also confusing as MSIMI is a very specific thing not clearly related to the prevention and treatments discussed. (Minor: there are much more to be said about psychological treatments and the reference used seem to have gotten the author order of the Cochrane reveiw wrong).

This ms does not describe (at least not much) the bidirectional relationship between mental stress and CVD. And it doesn't describe the behavioural mechanisms. Maybe that is Part I.

In summary, to my understanding, well written, very good content (impressing) but very condensed under unclear circumstanes and the context/aim/rational is lost on me.

In the right context, maybe a book chapter going through the physiological mechanisms, this could be really good. As an independent article, I'm not sure.

Author Response

Mental stress effect on cardiovascular health - Part II

Responses to Reviewers

We are thankful for the reviewers’ efforts in reviewing our manuscript and for the valuable suggestions which have strengthened it. We hereby address the raised comments.

Reviewer #1

Reviewer comment. My major concern is that I do not understand the context or rational. The title suggests "Part II". Where is then part I? It is some kind of ambitious but super-condensed literature review but the title is not informative and it doesn’t follow the normal structure for a review. Nor is there any aim or any guiding text for the reader to help him/her understand the context or the rational behind this article.

Authors response. The Introduction section has been re-written, explaining what is discussed in Part I (which has been submitted to JCM on 28 March, still awaiting decision), and the aim of Part II.

Reviewer comment. To my understanding the content in this ms is overall correct. However, it is not obvious what the principles for selection of the content are. Some headings are misleading. E.g. the Introduction is not an introduction (an introduction would have been appreciated). Mental Stress-Induced Cardiomyopathy is not the same thing as Takotsubo which is suggested by one heading. Putting MSIMI together with prevention and treatment is also confusing as MSIMI is a very specific thing not clearly related to the prevention and treatments discussed. (Minor: there are much more to be said about psychological treatments and the reference used seem to have gotten the author order of the Cochrane review wrong).

Authors response. About the misleading headings:

The Introduction has been re-written.

We agree that Mental stress-induced cardiomyopathy is not the same as Takotsubo. The heading has been changed accordingly. A short sentence has been added to indicate that few patients do not have any preceding stressful event.

MSIMI prevention and treatment: MSIMI is discussed in Part I. The heading has been changed.

The results of the Cochrane Review have been detailed:

A comprehensive systematic review of several studies of psychological treatments in patients with CAD, including people with and without a psychopathology at baseline, showed no effect on total mortality, myocardial revascularization procedures, or non-fatal myocardial infarction. However, there was low-quality evidence of small reduction of cardiac mortality and improvement of psychological symptoms, such as depression, anxiety or stress.

Reviewer comment. This ms does not describe (at least not much) the bidirectional relationship between mental stress and CVD. And it doesn't describe the behavioural mechanisms. Maybe that is Part I.

 Authors response. Thank you for the comment. The bidirectional relationship between mental stress and CAD is described in Part I. This point has been added to the Introduction.

Reviewer 2 Report

Dear Henein et al.,

The manuscript “Mental stress effect on cardiovascular health - Part II” (jcm-1689172) by Henein et al. summarize the correlation of mental stress on cardiovalscular health. The topic is interesting, but I think this article should reconsider after proper changes in major revision for publication in Journal of Clinical Medicine. Some of my specific comments are:

  1. In line 2 of the tittle in the present review article, what is the mear of “Part II”? Is there the part I of related review article ?
  2. Describe the novelty of the article made by the author? From the results of my evaluation, it seems that many similar published works adequately explain what you have raised in the current manuscript related to mental stress effect in relation with cardiovascular health as the best reviewer knowledge in this research area. If there is something others really new in this manuscript, please highlight it more clearly in the introduction section (line 37-46).
  3. State of the art and significance of the present article is not clearly present, the authors should highlight it more advanced in the introduction section (line 37-46).
  4. To support the introduction explanation, where nothing uses any references. I would encourage and advise the authors to adopt some of the specific additional references related to mental stress by MDPI in the introduction section as follow:
  • Effect of Short-Term Deep-Pressure Portable Seat on Behavioral and Biological Stress in Children with Autism Spectrum Disorders: A Pilot Study. Bioengineering 2022, 9, 48. https://doi.org/10.3390/bioengineering9020048
    • The Subjective Comfort Test of Autism Hug Machine Portable Seat. J. Intellect. Disabil. - Diagnosis Treat. 2021, 9, 182–8. https://doi.org/10.6000/2292-2598.2021.09.02.4
  1. To enhance the quality of present review, the authors should add one systematic figure/summary figure to illustrate the summary of current review to make the reader more interested and easier to understand rather than only using dominant text to explain as presented in the author’s manuscript.
  2. The authors should add at least one paragraph to describe the limitation of the review carried out that would be added before conclusion section (before line 318).
  3. The conclusion of the present manuscript is not solid. Further elaboration is needed (line 318-326).
  4. Further research needs to be explained in the conclusion section (line 318-326).
  5. To improve the quality of English used in this manuscript and make sure English language, grammar, punctuation, spelling, and overall style are correct, further proofreading is needed. As an alternative, the authors can use the MDPI English proofreading service for this issue.
  6. Please make sure the authors have used the Journal of Clinical Medicine, MDPI format correctly. The authors can download published manuscripts by Journal of Clinical Medicine, MDPI, and compare them with the present author's manuscript to ensure typesetting is appropriate. Some errors are:
  • Thre is a ted text on line 39 that shoul be changed into black colour
  • Information about Authors Contribution, Funding, Institutional Review Board Statement, Informed Consent Statement, Data Availability Statement, Acknowledgment, and Conflict of Interest should be put after conclusion section and before references.
  • And others

I am pleased to have been able to review the author's present manuscript. Hopefully, the author can revise the current manuscript as well as possible so that it becomes even better. Good luck for the author's work and effort.

Best regards,

The Reviewer

Author Response

Mental stress effect on cardiovascular health - Part II

Responses to Reviewers

We are thankful for the reviewers’ efforts in reviewing our manuscript and for the valuable suggestions which have strengthened it. We hereby address the raised comments.

Reviewer #2

  1. Reviewer comment. In line 2 of the tittle in the present review article, what is the mear of “Part II”? Is there the part I of related review article ?

 Authors response. There is a new Introduction explaining that The role of central nervous system (CNS) in the appraisal of environmental stimuli, translating negative emotions into hemodynamic and vascular changes, as well as their effects on myocardial function, and the bidirectional relationship between mental stress and CAD, are discussed in Part I (submitted to JCM on 28 March 2022).”

  1. Reviewer comment. Describe the novelty of the article made by the author? From the results of my evaluation, it seems that many similar published works adequately explain what you have raised in the current manuscript related to mental stress effect in relation with cardiovascular health as the best reviewer knowledge in this research area. If there is something others really new in this manuscript, please highlight it more clearly in the introduction section (line 37-46).

Authors response. This review is not a research paper, proposing new hypotheses or results. However, it is based on most recent literature, se stated in the Introduction “The aim of Part II is to review the evidence for an association between mental stress and systemic CV effects, based on most recent literature findings.”

  1. Reviewer comment. State of the art and significance of the present article is not clearly present, the authors should highlight it more advanced in the introduction section (line 37-46).

Authors response. The Introduction has been revised according to the indications of the Reviewers

  1. Reviewer comment. To support the introduction explanation, where nothing uses any references. I would encourage and advise the authors to adopt some of the specific additional references related to mental stress by MDPI in the introduction section as follow:
  • Effect of Short-Term Deep-Pressure Portable Seat on Behavioral and Biological Stress in Children with Autism Spectrum Disorders: A Pilot Study. Bioengineering 2022, 9, 48. https://doi.org/10.3390/bioengineering9020048
    • The Subjective Comfort Test of Autism Hug Machine Portable Seat. J. Intellect. - Diagnosis Treat. 2021, 9, 182–8. https://doi.org/10.6000/2292-2598.2021.09.02.4

Authors response. We thank the Reviewer for the interesting references suggestion. However, although these studies demonstrate a reduction in neurobiological stress with the use of autism hug machine portable seat, they do not have any relationship with cardiovascular disease. Some other relevant references have been added to the Introduction.

  1. Reviewer comment. To enhance the quality of present review, the authors should add one systematic figure/summary figure to illustrate the summary of current review to make the reader more interested and easier to understand rather than only using dominant text to explain as presented in the author’s manuscript.

Authors response. A summary Figure 1, illustrating the relationship between mental stress and systemic cardiovascular effects, has been added to “Inflammatory response to mental stress”.

  1. Reviewer comment. The authors should add at least one paragraph to describe the limitation of the review carried out that would be added before conclusion section (before line 318).

Authors response. “Limitations” paragraph has been added before Conclusion.

7 & 8. Reviewer comment. The conclusion of the present manuscript is not solid. Further elaboration is needed (line 318-326).

Further research needs to be explained in the conclusion section (line 318-326).

Authors response. Some changes have been added to the Conclusion.

  1. Reviewer comment. To improve the quality of English used in this manuscript and make sure English language, grammar, punctuation, spelling, and overall style are correct, further proofreading is needed. As an alternative, the authors can use the MDPI English proofreading service for this issue.

Authors response. The text has been extensively revised by one of the author (MYH), native English language speaker.

  1. Reviewer comment. Please make sure the authors have used the Journal of Clinical Medicine, MDPI format correctly. The authors can download published manuscripts by Journal of Clinical Medicine, MDPI, and compare them with the present author's manuscript to ensure typesetting is appropriate.Authors response. Thank you. This was already done in the submission process.

Reviewer 3 Report

Although this topic is really complex and much is not known, the manuscript is overall well written and well balanced. This is a nice review paper. I have nothing to comment.

Round 2

Reviewer 1 Report

The authors have revised according to my main comments (but still reference 145 has erroneous author order). The format is still a bit unortodox. Is it a review or not? It doesn't follow any Equator guideline. It reads more like a chapter in a coursebook. On the other hand I find the content correct although not novel. There are other articles summarizing similar topics. If part 1 is accepted, maybe this one will be a nice complement.

Reviewer 2 Report

Dear Henein et al.,

After carefully reading the author's revised manuscript entitled "Mental stress effect on cardiovascular health - Part II" (jcm-1689172) by Henein et al., with all due respect, I recommend the manuscript must be rejected for publication on Journal of Clinical Medicine. Regarding the review being carried out on the current manuscript, there is a misunderstanding that is very dangerous for the scientific community.

This manuscript is so minimal in substance and not existence scientific contribution that it is not suitable to be published in a separated manuscript between Part I and Part II, the author should have made it into one complete manuscript when he wanted to publish it. Even Part II manuscript by this authors is published after my rejection decision, it will have the effect of biasing unworthy science.

In the end, this manuscript is inappropriate for publication, has a serious flaw, and should not be published.

Best regards,

The Reviewer